# Retrospective evaluation of a novel ultrasound-based imaging analysis software for predicting radiofrequency ablation areas

Masaya Sato [1,2], Ryosuke Tateishi [2]*, Yogev Zohar[3], Jiro Sato[4], Takeyuki Watadani[5], Makoto Moriyama[2], Taijiro Wake[2], Ryo Nakagomi[2], Mizuki Nishibatake Kinoshita[2], Takuma Nakatsuka[2], Tatsuya Minami[2], Koji Uchino[2], Kenichiro Enooku[2], Hayato Nakagawa[2,6], Yoshinari Asaoka [2,7], Ryo Yamada[8], Nitzan Even[3], Inbal Amitai[3], Yossi Abu[3], Mitsuhiro Fujishiro [2], Kazuhiko Koike[2]

1 Department of Clinical Laboratory Medicine, Graduate School of Medicine, The University of Tokyo, Tokyo, Japan, 2 Department of Gastroenterology, Graduate School of Medicine, The University of Tokyo, Tokyo, Japan, 3 TechsoMed Medical Technologies Ltd., Rehovot, Israel, 4 Department of Radiology, Tokyo Metropolitan Police Hospital, Tokyo, Japan, 5 Department of Radiology, The University of Tokyo, Tokyo, Japan, 6 Department of Gastroenterology and Hepatology Mie University Graduate School of Medicine, Mie, Japan, 7 Department of Medicine, Teikyo University School of Medicine, Tokyo, Japan, 8 Development Department 2, SCREEN Advanced System Solutions Co., Ltd, Kyoto, Japan

* tateishi-tky@umin.ac.jp

**Data Availability Statement:** All relevant data are within the paper and its Supporting Information files.

## Abstract

### Objective

This study aimed to introduce and evaluate a novel software-based system, BioTrace, designed for real-time monitoring of thermal ablation tissue damage during image-guided radiofrequency ablation for hepatocellular carcinoma (HCC).

### Methods

BioTrace utilizes a proprietary algorithm to analyze the temporo-spatial behavior of thermal gas bubble activity during ablation, as seen in conventional B-mode ultrasound imaging. Its predictive accuracy was assessed by comparing the ablation zones it predicted with those annotated by radiologists using contrast-enhanced computed tomography (CECT) 24 hours post-treatment, considered the gold standard. The study included 20 liver tumors.

### Results

The median tumor measurement along the major axis was 1.2 cm. The median Dice coefficient, Sensitivity, and Precision between BioTrace and CECT were 0.90, 0.91, and 0.91, respectively. The intraclass correlation showed excellent agreement in volume size between BioTrace and CECT findings (0.98).

### Conclusion

BioTrace effectively generates an ablation damage prediction map based on real-time ultrasound imaging, accurately predicting the ablation zone as confirmed by 24-hour post-

**Funding:** This work was supported by the Health, Labor, and Welfare Policy Research Grants from the Ministry of Health, Labor, and Welfare of Japan (Policy Research for Hepatitis Measures [H30-Kansei-Shitei-003]).

**Competing interests:** This study was funded by TechsoMed Medical Technologies and designed in collaboration with the University of Tokyo and SCREEN Holdings Co., Ltd. RT has received lecture fee from TechsoMed Medical Technologies. RY is an employee of SCREEN Holdings Col, Ltd and a shareholder in the company. RY provided technical support for the BioTrace system. YZ is an employee of TechsoMed Medical Technologies. YZ involved in study design, data analysis, and drafting of the manuscript. NE is an employee of TechsoMed Medical Technologies. NE involved in drafting of the manuscript. IA is an employee of TechsoMed Medical Technologies. IA involved in data analysis. YA is an employee of TechsoMed Medical Technologies and a shareholder in the company. YA involved in study design and drafting of the manuscript. This does not alter our adherence to PLOS ONE policies on sharing data and materials.

**Abbreviations:** B-mode, brightness-mode; CI, confidence interval; CT, computed tomography; FN, false negative; FP, false positive; HCC, hepatocellular carcinoma; HDMI, high-definition multimedia interface; IBS, integrated backscatter; ICC, intraclass correlation coefficient; MRI, magnetic resonance imaging; PC, personal computer; RFA, radiofrequency ablation; TN, true negative; TP, true positive; TVI, tissue viability imaging; VGA, video graphics array.

procedural CECT. This system has the potential to enhance the safety and efficacy of ablation procedures in clinical settings.

## Introduction

Image-guided minimally invasive techniques, including radiofrequency ablation (RFA), have become essential in treating hepatocellular carcinoma (HCC), particularly for patients who are not candidates for surgical resection due to impaired hepatic function or associated comorbidities [1–3]. While RFA offers promising clinical results, comparable to surgical resection, it presents challenges in achieving precise ablation margins using ultrasound guidance alone [4]. This limitation often necessitates additional imaging modalities like computed tomography (CT), magnetic resonance imaging; (MRI), and contrast-enhanced ultrasonography post-RFA to assess treatment success.

A significant issue with RFA is the potential for incomplete ablation, requiring multiple treatment sessions. This often arises from irregular-shaped burns caused by the cooling effect of nearby large vessels [4]. Real-time monitoring during the ablation process can help operators identify incomplete ablation and adjust the procedure immediately, potentially reducing the need for multiple sessions and minimizing complications.

To address this need, we present BioTrace (TechsoMed Medical Technologies, Israel), an innovative software designed to generate ablation damage maps in real time (BioTrace map or BTM) using brightness-mode (B-mode) ultrasound imaging (Fig 1). BioTrace analyzes echogenicity variations caused by the temporo-spatial behavior of local gas bubble activity and microvascular changes during thermal ablation. By tracking and classifying these variations, BioTrace estimates the extent of thermal damage at 24 hours post-treatment, providing real-time feedback to physicians. This will enable adjustments during the ablation process, ensuring sufficient margins for complete tumor ablation while minimizing damage to surrounding tissues.

This study aims to evaluate the performance of BioTrace in predicting the ablation area in real-time during RFA for liver tumors in clinical settings.

## Methods

### Patients and procedures

This study is a retrospective analysis of prospectively collected ultrasound imaging data, aiming to evaluate the predictive capabilities of the BioTrace system. The study protocol conformed to the ethical guideline of the Declaration of Helsinki and was approved by the University of Tokyo Medical Research Ethics Committee (approval number 11499). We prospectively enrolled patients who underwent RFA for HCC at our institution between July 2017 and November 2018. The inclusion criteria were (1) single ablation procedure not requiring one or more additional ablation procedures for a single tumor and (2) ablation using the B-mode ultrasonography guidance (ablation under the guidance of contrast-enhanced ultrasonography was not included). HCC lesions included in this study were undergoing initial treatment. All patients provided written informed consent. All RFA procedures were performed using either Medtronic Cool Tip RF generator and Cool-Tip electrode (Cool-tip™ RF Ablation, Medtronic, Minneapolis, MN, USA) or STARmed VIVA generator and VIVA 17 Gauge RF electrode (VIVA, STARmed, Goyang, South Korea) under ultrasonic guidance using a LOGIQ

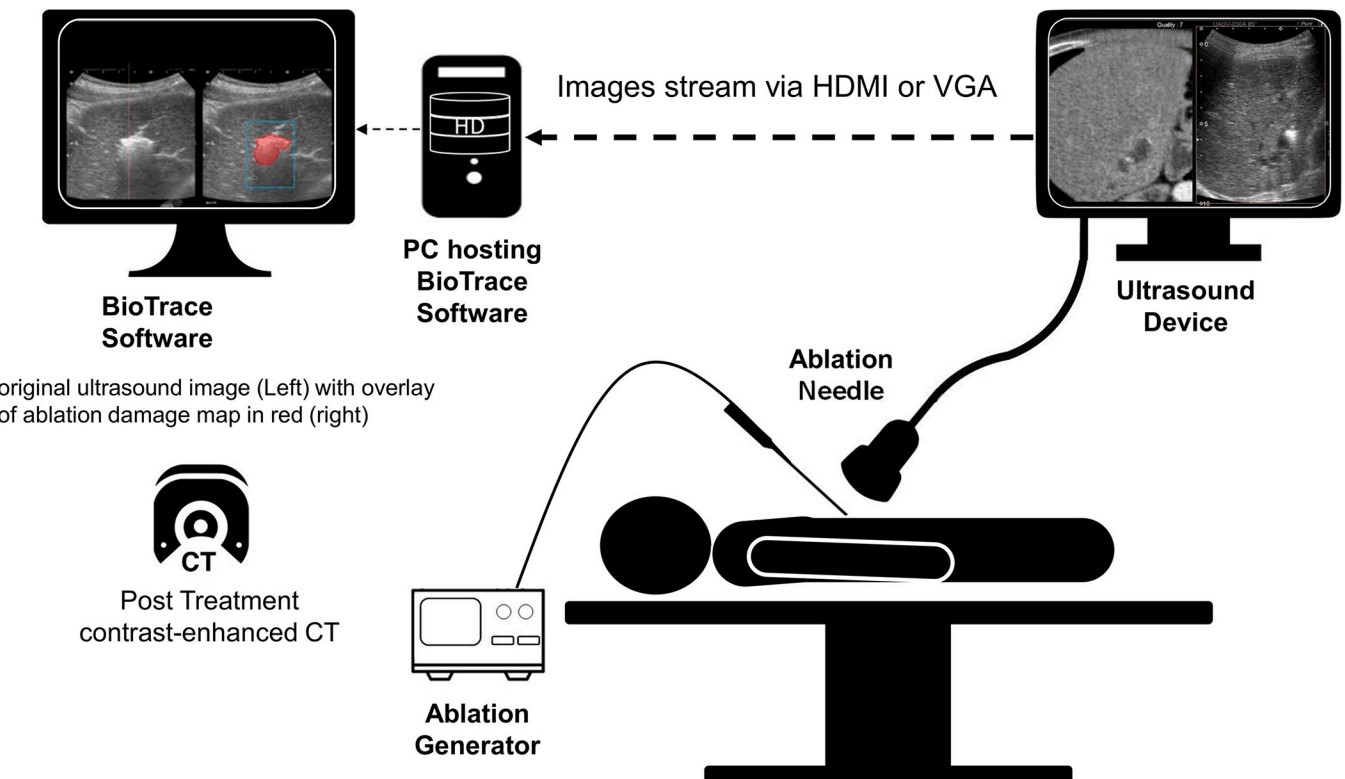

**Fig 1. Schematic illustration of the use of BioTrace in clinical settings and the information flow as used in online mode.** CT, computed tomography; HDMI, high-definition multimedia interface; PC, personal computer; VGA, video graphics array.

E9 (GE Healthcare, Chicago, IL, USA) and Toshiba Aplio 500 instrument (Canon Medical Systems Co., Tokyo, Japan).

## Outline of online monitoring system

We validated a novel imaging analysis software, BioTrace™ (TechsoMed Medical Technologies, Israel), which is based on observing and analyzing the temporospatial behavior of the local thermal microbubble activity spontaneously generated by thermal treatment without the use of contrast agents. By tracking the microbubbles activity, the BioTrace algorithm recognizes the local changes in tissue viability online during the entire ablation process (Fig 1). The phenomenon of heat based microbubbles formation, also known as the 'gas bubble effect' [5], was previously reported during both RF and microwave (MW) ablations [6, 7].

The dynamics of microbubbles reflect a complex interaction of multiple variables during thermal ablation. While temperature is a primary factor, the microbubble behavior is also influenced by: (1) the biological nature of the lesion, (2) the underlying pathological condition of the liver (such as chronic hepatitis, cirrhosis, or metabolic liver disease), (3) the degree of liver fibrosis, (4) local tissue perfusion characteristics, and (5) the applied temperature. These variables collectively affect the formation and the dynamic behavior of microbubbles during thermal ablation. Temperatures exceeding 60˚C cause rapid protein denaturation and complete melting of the plasma membrane adding to nitrogen release and microbubble formation. A further rise in temperature is associated with physical changes such as water vaporization, desiccation, and carbonization [8]. When tissue temperature further increases and reaches 100˚C (as occurs with RF or MW during thermal liver ablation [9]), microbubbles continue to

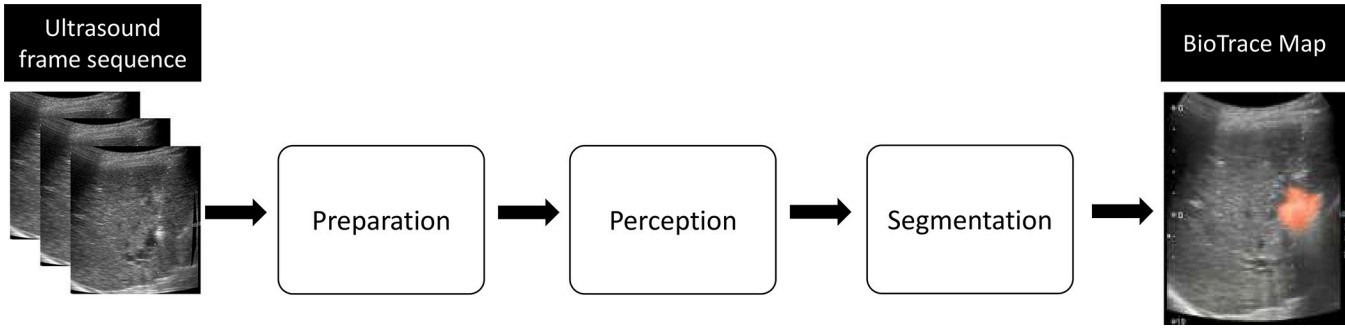

**Fig 2. Schematic description of the three-phased principle of operation of the BioTrace.**

form due to tissue boiling. The BioTrace algorithm tracks the echogenicity changes induced by the formed microbubbles and their dynamics, together with classification of micro vascularity changes. These echogenicity variations can be used to define several criteria associated with damage, which, once met, can be used to identify ablation tissue damage.

The BioTrace algorithm analyzes frame sequences at pre-set time points, acquired from the ultrasound system during the ablation procedure using three-phased process (Fig 2) to produce an ablation damage map (BioTrace Map) displayed as a cover overlay map on the gray-scale B-mode image (S1 Fig). The preparation phase filters poor image quality and "out of plane" input frames using image quality assessment and registration. The second perception phase is designed to extract features and use them in preparation for the third and final segmentation phase that enables the calculation of the evolving tissue damage and its presentation as BioTrace map. Using this process, the algorithm is designed to overcome real-world variables such as breathing movements and shadowed areas (rib and ablation shadows) which are characterized by lower signal to noise ratio while tracking and visualizing the ablation damage over real-time ultrasound imaging.

## Performance evaluation of BioTrace and statistical analysis

In this study, we retrospectively analyzed the ablation areas by comparing the BioTrace prediction generated from ultrasound imaging data collected during the RFA procedure with the actual ablation zones confirmed by CECT obtained 24 hours post-treatment, which is considered the gold standard. While the BTM system is designed for real-time monitoring, it was not activated during the procedures in this study. Instead, we retrospectively applied the BioTrace algorithm to the prospectively collected ultrasound imaging videos to generate the ablation prediction maps. In order to assess the predictive accuracy of BioTrace monitoring, we used the Dice coefficient criterion [10] as well as Sensitivity and Precision metrics.

The results were described as true positive, if both BioTrace and the radiologists marked the tissue as ablated; true negative, if BioTrace and radiologists marked the tissue as un-ablated; false positive, if BioTrace marked the tissue as ablated but the radiologists determined it as un-ablated; and false negative, if both BioTrace marked the tissue as un-ablated but the radiologists marked the tissue as ablated (S2 Fig).

Since CECT is a 3D multi-plane scan and the map of the ablated area demonstrated by the BioTrace Map is depicted on 2D ultrasound plane, it was mandatory to register the precise post-interventional CECT 2D slice to the ultrasound field of view monitored for the BioTrace assessment. A multi-step registration and validation process was approved by radiologists and carried out as part of the results analysis. First, synchronization and registration (in real-time) of ultrasound images (transducer position) with multiplanes reconstruction pre-interventional

CECT/MRI image was carried out using commercially available volume navigation fusion imaging tools (Smart fusion software [Canon Medical Systems Co., Tokyo, Japan] and volume navigation [V-nav] software [GE Healthcare, Chicago, IL, USA]). The navigation system synchronizes real-time US images with CT images using plane and point registration to achieve optimal fusion overlay of the target lesion. Then, a rigid registration of the CECT/MRI diagnostic to the 24-h CECT scan was carried out based on anatomical structures (i.e anatomical features in the vicinity of the target tissue such as hepatic and portal veins, liver edge, vena cava). A Multi-Planar Reconstruction (MPR) was used to transform the axial plane images into the fused CT-US view (transducer procedure position), also defined as "ablation oblique plane". The BioTrace Map is then registered to an ultrasound calibration image captured during the procedure and the BioTrace system calculates the statistical metrics-comparison of Bio-Trace Map to the segmented ablated area on the 24-hh CECT (as defined by the physicians).

The target registration error of the ultrasound-CT fusion system was previously estimated to be roughly 5mm [11]. In this study, in order to compensate for the error introduced by the fusion-navigation system and in order to achieve registration accuracy acceptable by the physician, a manual registration correction tool was implemented. This tool enabled the physician to manually correct any translation, scaling or rotation errors, thus ensuring that the CT and ultrasound images are aligned for user satisfaction.

The BioTrace predictive ablation map was determined after the completion of the ablation procedure based on the ultrasound image acquired in real-time from the same ultrasound plane throughout the procedure. In order to make sure we maintain the same ultrasound plane throughout the procedure, the ultrasound probe was kept using a bracket (biopsy guide) that bundled mechanically between the ultrasound transducer and the ablation needle. As a result, we were able to carry out the analysis on a single plane and eliminate potential errors due to change of planes. The BioTrace system also includes ultrasound to ultrasound frames registration to compensate for patient breathing, transducer movement and out of plane frames filtration.

We evaluated the reproducibility of the depiction of the ablation areas of BioTrace and those of the radiologists using intraclass correlation coefficient (ICC) and Bland–Altman limit of agreement [12]. The Pearson's correlation coefficient was used to examine the linear relationship to evaluate the proportional bias of the Bland–Altman plot. Concordance, Sensitivity, and Precision were interpreted to have poor agreement (<0.7), fair agreement (0.7–0.8), good agreement (0.8–0.9), or excellent agreement (0.9–1.0) [13]. Moreover, the ICC was interpreted as minimally acceptable (0.7–0.8), good (0.8–0.9), or excellent (0.9–1.0) [14]. To address the differences in efficacy of the BioTrace map across different liver segments, we employed an analysis of variance (ANOVA) test. Due to the exploratory nature of this study, each experiment involving the BioTrace system was conducted once. Statistical analyses were performed using R software, version 3.4.3 (https://cran.r-project.org/).

## Results

### Patient and tumor characteristics

We evaluated the performance of BioTrace in evaluating the ablated area for 20 liver tumors. Table 1 shows the patient and tumor characteristics. All tumors were HCCs, measuring ≤2.0 cm in the major axis. The median (interquartile range) measurement in the major axis of the tumors was 1.2 (0.98–1.55) cm.

Continuous variables are presented as the medians (1st-3rd quartile), while categorical variables are presented as the number and frequency (%).

**Table 1. Patient and tumor characteristics.**

| Parameter | Values |
|---|---|
| Age (years) | 72 (64–76) |
| Male sex | 8/13 (61.5%) |
| Tumor size | 1.2 (1.0–1.5) |
| Tumor location | |
| Segment 1 | 0/20 (0%) |
| Segment 2 | 0/20 (0%) |
| Segment 3 | 2/20 (10.0%) |
| Segment 4 | 0/20 (0%) |
| Segment 5 | 3/20 (15.0%) |
| Segment 6 | 5/20 (25.0%) |
| Segment 7 | 5/20 (25.0%) |
| Segment 8 | 5/20 (25.0%) |

## Predictive accuracy of BioTrace Map for demonstrating the ablated area

Table 2 and S3 Fig show the comparison between the ablated areas demonstrated by BioTrace Map and by the radiologists based on contrast-enhanced CT (CECT) obtained 24h after treatment. The ablated areas prediction demonstrated by BioTrace and the radiologists are shown in red and blue color maps, respectively (S3 Fig). The median and interquartile range values of the Dice coefficient, Sensitivity, and Precision between BioTrace and the 24-h CECT ablation zone reached 0.90 (0.88–0.92), 0.91 (0.89–0.94), and 0.91 (0.86–0.94), respectively.

**Table 2. Ablated areas demonstrated by BioTrace and the radiologists.**

| Tumor number | Tumor segment | BioTrace cross-section surface area (mm$^2$) | Radiologists Cross-section surface area (mm$^2$) | Dice coefficient (%) | Sensitivity (%) | Precision (%) |
|---|---|---|---|---|---|---|
| 1. | 6 | 477.66 | 470.67 | 94 | 93 | 94 |
| 2. | 7 | 555.70 | 556.85 | 88 | 88 | 88 |
| 3. | 5 | 687.91 | 687.10 | 93 | 92 | 93 |
| 4. | 8 | 475.08 | 536.71 | 90 | 96 | 85 |
| 5. | 7 | 423.36 | 362.47 | 87 | 81 | 94 |
| 6. | 3 | 874.05 | 871.99 | 89 | 89 | 89 |
| 7. | 7 | 404.34 | 362.62 | 87 | 83 | 92 |
| 8. | 6 | 964.98 | 1032.03 | 89 | 92 | 86 |
| 9. | 8 | 617.86 | 628.99 | 94 | 94 | 93 |
| 10. | 8 | 282.36 | 307.16 | 88 | 94 | 81 |
| 11. | 6 | 248.09 | 259.17 | 92 | 89 | 94 |
| 12. | 5 | 362.60 | 326.69 | 91 | 86 | 96 |
| 13. | 5 | 333.11 | 338.43 | 91 | 91 | 90 |
| 14. | 6 | 268.69 | 253.06 | 93 | 95 | 91 |
| 15. | 7 | 448.92 | 498.74 | 90 | 95 | 85 |
| 16. | 8 | 255.36 | 296.56 | 89 | 94 | 84 |
| 17. | 7 | 777.05 | 723.26 | 93 | 90 | 96 |
| 18. | 8 | 599.99 | 524.73 | 88 | 82 | 94 |
| 19. | 3 | 440.57 | 439.80 | 91 | 91 | 91 |
| 20. | 6 | 726.21 | 773.61 | 88 | 91 | 85 |

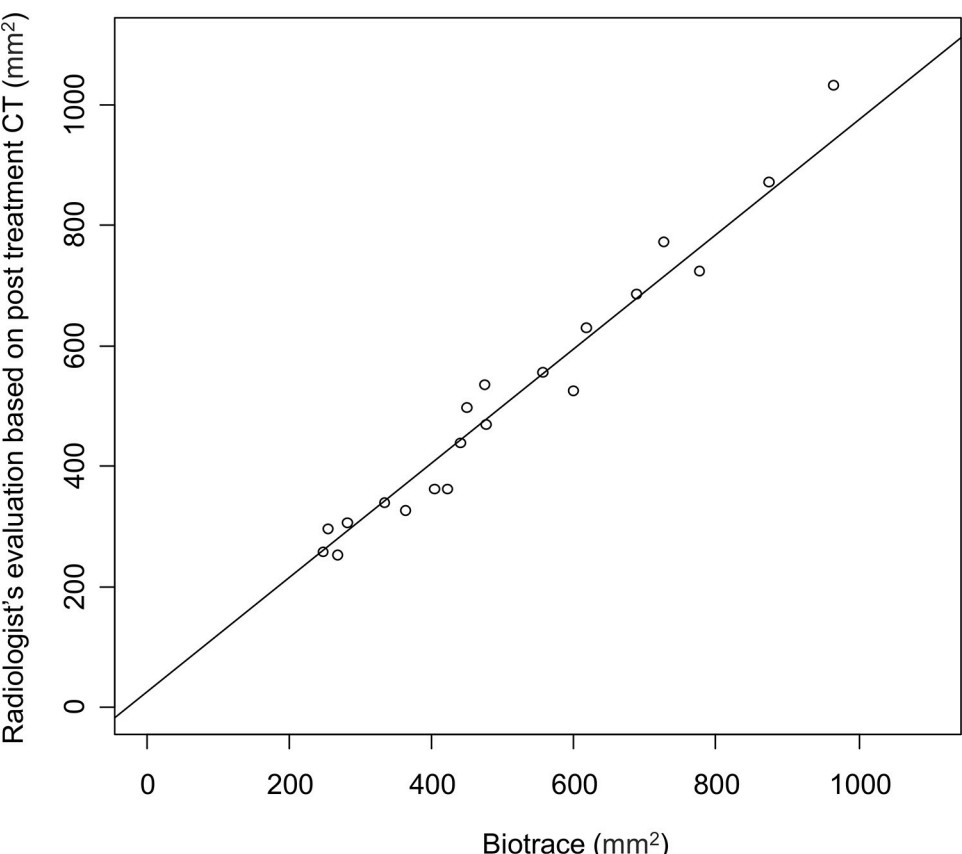

**Fig 3. The dot-plot constructed based on the relationship between the sizes of the ablated areas described by BioTrace and the radiologist's evaluation based on post treatment CT.**

We also compared the cross-section area reproducibility of the ablated areas between Bio-Trace and the radiologists. Fig 3 shows the correlation between the sizes of the ablated areas described by BioTrace and the radiologists. Excellent agreement on cross-section area was observed (ICC, 0.98; 95% confidence interval [CI], 0.96–0.99). Fig 4 shows the Bland–Altman plot constructed by assigning the mean measurements of the ablated areas by the radiologists and BioTrace as the X-axis and the difference between the two measurements as the Y-axis. No evidence of systematic bias was observed with the intercept (95% CI) of the Bland-Altman plot of −1.34 (−80.7 to 78.0). Moreover, the Pearson's correlation coefficient showed no evidence of proportional bias (correlation = −0.17; p = 0.45) (Fig 5). The ANOVA test did not reveal differences based on the tumor's localized segment (p = 0.4).

## Discussion

Although ultrasound imaging offers a real-time, cross-sectional image to help the guidance of the RFA needle, the image itself offers limited treatment capability, as it tracks only the changes in the image brightness or tissue echogenicity. Various methods to track the ultrasound echogenicity changes (i.e, deformation or decorrelation imaging) have been reported for the development of a monitoring system to visualize the ablated area during ablation procedures [15–18]. Echo decorrelation imaging is a pulse-echo method that maps heat-induced changes in ultrasound imaging. Subramanian et al. investigated the correlation between echo decorrelation coefficient and temperature by simulating the temperature field during the RFA

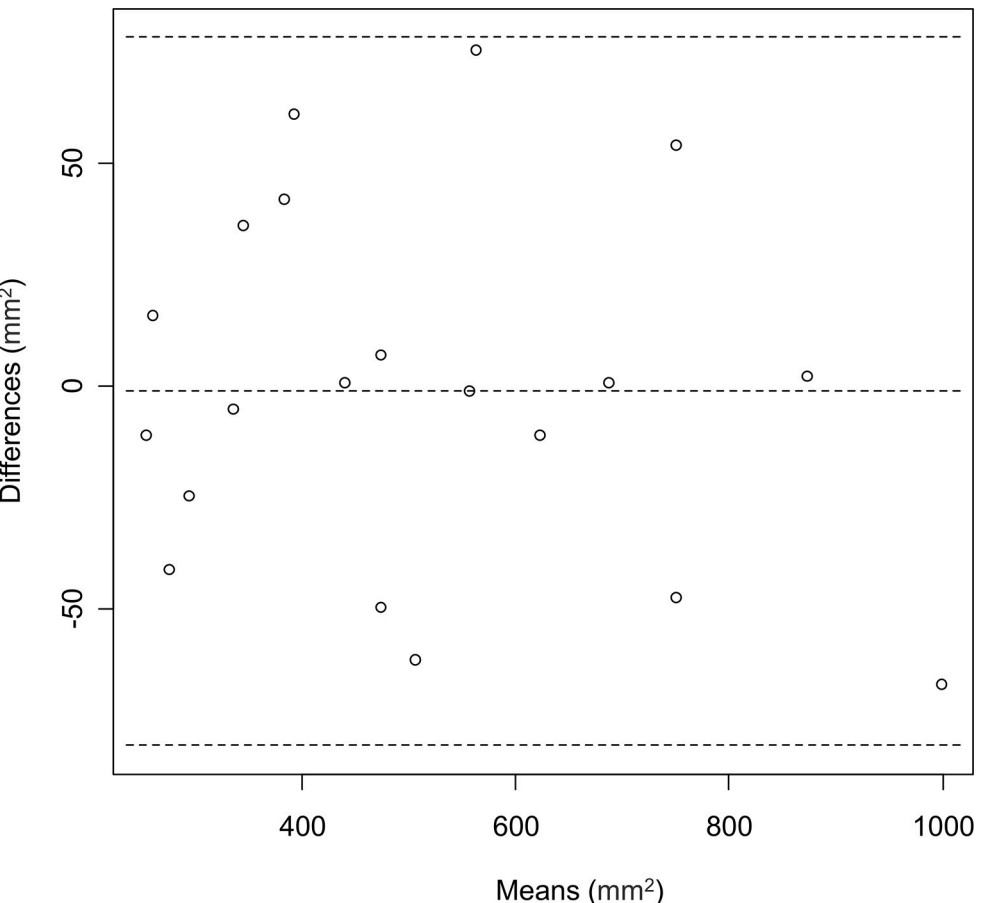

**Fig 4. Bland–Altman plot constructed by assigning the mean measurements (x-axis) of the ablated areas by the radiologists and BioTrace and the difference between the measurements (y-axis).**

of the liver ex vivo and verified the potential of ultrasound decorrelation imaging in thermal ablation monitoring [19]. Other studies have investigated tissue changes or motion and noise artifacts in echo decorrelation imaging in animal liver cancer models and demonstrated the potential of echo decorrelation imaging for quantifying the tissue damage during ablation [20, 21]. However, this method is still challenging, especially in vivo, because of the potential decorrelation artifacts caused by respiratory motion, cardiac motion, or perfusion [22]. In this study, we presented a novel real-time monitoring system based on microbubble activity using B-mode imaging for ablated areas to overcome the problem of echo decorrelation imaging.

Dice coefficient, Sensitivity, and Precision values showed excellent agreement between the Bio-Trace Map and CECT obtained 24-h post-RFA, which is considered the gold standard in ablation area evaluation (0.90, 0.91, and 0.91, respectively). Excellent reproducibility of BioTrace as compared to the gold standard was also observed when assessing the ablation cross-section area, without any evidence of systematic or proportional biases based on the Bland–Altman plot.

The BioTrace system offers several distinct advantages over contrast-enhanced ultrasonography. First, it minimizes the resources required for contrast agent preparation and administration. Second, unlike CEUS which requires low acoustic pressure to maintain microbubble stability, BioTrace can utilize optimal ultrasound settings to maintain high-quality B-mode imaging throughout the procedure. This allows for better visualization of the target area and surrounding structures during ablation.

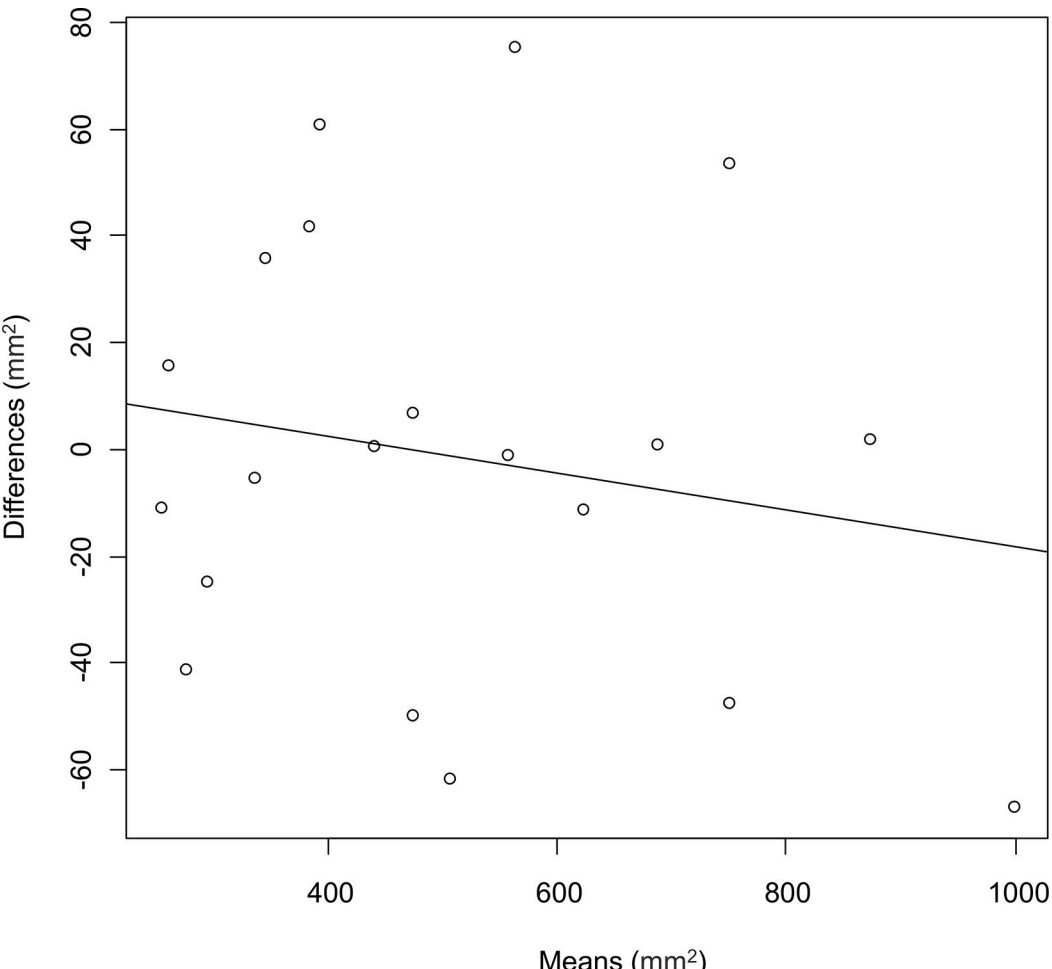

**Fig 5. The dot-plot constructed based on the relationship between the mean measurements of the ablated areas by the radiologists and BioTrace and difference between the measurements.**

The automatic suggestion to conduct ablation for unnoticed and insufficiently ablated areas due to the cooling effect of the large vessels [4] may help to avoid additional treatment sessions in patients, which is associated with a risk of complications or extended hospitalization. Moreover, the development of an alerting system to warn the operators regarding the involvement of important surrounding structures during ablation based on the current system will ensure safer RFA treatment with fewer complications. Although the safety margin was reported to be overestimated on axial CT images [23], CT is widely used for the safety margin assessment of RFA [24]. In clinical practice, the BioTrace map is expected to be used in real time during ablation to confirm the adequacy of the margins. This approach ensures that sufficient tissue has been ablated and helps identify any areas needing further treatment. Additionally, the Bio-Trace system demonstrated comparable performance to CT in assessing ablation zones in this retrospective analysis, suggesting potential advantages for future intra-operative monitoring applications. This capability could lead to a reduction in patient radiation exposure in the future, enhancing the safety and effectiveness of the procedure.

Previously reported pathology evaluation showed that it takes at least 12 hours after ablation for the damaged area to stabilize in its general final contour [25, 26]. A gradual increase in the ablation zone with stabilization of the damaged area was demonstrated to occur close to

24-h post ablation due to delayed pathological cellular and tissue injuries. The phenomenon of delayed apoptosis due to ischemia is well documented in scientific literature [27], indicating that around 24-h post ablation, most of the secondary damage to the tissue has evolved. A recent prospective imaging study performed at Mayo Clinic, AZ, USA, used the BioTrace software to compare the change in ablation zone volume between two post-operative CECT scans: immediate and 24h post operation for 26 liver tumor cases treated with percutaneous MW ablation [28]. A consistent volume expansion was noted 24h post ablation with median volume expansion of 57%. This phenomenon is also supported by retrospective thermal ablation imaging studies showing an increase of the mean diameter of the necrotic zone on 24-h post procedural imaging vs. imaging acquired immediately post procedure [29, 30]. Therefore, it can be stated that the ablated area seen on the CECT acquired 24-h. post procedure, reflects both the acute real-time tissue damage driven directly by the heat introduction during the procedure as well as from the delayed thermal effect. By tracking microbubble behavior and micro vascularity changes, the BioTrace Map reflects the entire area of tissue damage resulting from both primary and delayed thermal effects.

The limitation of this study is its small sample size of 20 cases, which may not be sufficient to draw definitive and robust conclusions regarding the efficacy and accuracy of the BioTrace system. Furthermore, we included only those cases treated in a single ablation session. Multiple ablation sessions are often required to obtain sufficient ablation area in daily clinical practice. In cases of re-ablation due to insufficient ablation, BioTrace may be used to analyze the new gas bubble activity and to generate an ablation map in real time, allowing us to visualize the extended ablation area and assess the effectiveness of the repeated ablation. However, further investigation is needed to verify these observations and fully understand their implications in clinical practice. Furthermore, the influence of various liver pathologies and tissue characteristics on microbubble dynamics requires additional investigation. Future studies should evaluate the system's performance across different pathological conditions, including chronic hepatitis, cirrhosis, and varying degrees of liver fibrosis, to better understand how these factors affect the accuracy of ablation zone prediction. While the BioTrace system is designed for real-time monitoring, this validation study focused on retrospective analysis of prospectively collected ultrasound imaging data. The real-time monitoring capabilities will be evaluated in future prospective clinical trials. Additionally, the study did not involve any follow-up procedures, which limits our ability to assess the long-term impact and outcomes of the BioTrace system. Finally, the study did not include a traditional control group, which limits the ability to demonstrate the added clinical value of the BioTrace system. Further studies are needed to assess the potential clinical benefits and added value of the BioTrace system in a controlled setting.

In conclusion, we presented a novel ultrasound-based ablation monitoring system (BioTrace) designed for real-time visualization of the ablated area in liver tumor ablation. BioTrace effectively generated ablation damage prediction map based on real-time ultrasound imaging, offering accurate prediction of the ablation zone as visualized on 24-hour post-interventional CECT. This system may provide perspectives for safer and more effective ablation procedures in the future. Accessing predictions of ablation therapy outcomes in real-time may ultimately assist physicians in reducing complications, enhancing target coverage, and mitigating local tumor progression.

## Supporting information

**S1 Fig. Time evolution in consecutive ultrasound frames of BioTrace ablation damage map (red stains) as it forms during the ablation treatment.**
(TIF)

**S2 Fig. Conceptual diagram showing true positive, true negative, false positive, and false negative results of the ablated area described by BioTrace.**
(TIF)

**S3 Fig. BioTrace showing interface, showing a case from a clinical evaluation–comparing BioTrace ablation damage map (in red) with necrosis area as can be seen in 24h post CECT (in blue).**
(TIF)

**S1 File.**
(DOCX)

## Acknowledgments

This document has been edited by a professional English language editor, who is a native English speaker. We would like to thank Editage (www.editage.com) for English language editing.

## Author Contributions

**Conceptualization:** Masaya Sato, Ryosuke Tateishi, Yogev Zohar, Yossi Abu.

**Data curation:** Yogev Zohar, Makoto Moriyama, Taijiro Wake, Ryo Nakagomi, Mizuki Nishibatake Kinoshita, Takuma Nakatsuka, Tatsuya Minami, Koji Uchino, Kenichiro Enooku, Hayato Nakagawa, Yoshinari Asaoka.

**Formal analysis:** Ryosuke Tateishi, Yogev Zohar.

**Funding acquisition:** Kazuhiko Koike.

**Investigation:** Masaya Sato, Ryosuke Tateishi, Jiro Sato, Takeyuki Watadani, Nitzan Even, Inbal Amitai.

**Methodology:** Yossi Abu.

**Resources:** Ryo Yamada.

**Supervision:** Yossi Abu, Mitsuhiro Fujishiro, Kazuhiko Koike.

**Validation:** Jiro Sato, Takeyuki Watadani.

**Writing – original draft:** Masaya Sato.

**Writing – review & editing:** Ryosuke Tateishi, Yogev Zohar, Nitzan Even, Inbal Amitai, Yossi Abu, Mitsuhiro Fujishiro, Kazuhiko Koike.

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
