## [Decision Letter · Decision Letter 0]

11 Oct 2024

PONE-D-24-37385A Novel Real-Time Ultrasound-Based Imaging Analysis Software for Monitoring Radiofrequency Ablation and Visualizing Ablation AreasPLOS ONE

Dear Dr. Sato,

Thank you for submitting your manuscript to PLOS ONE. After careful consideration, we feel that it has merit but does not fully meet PLOS ONE’s publication criteria as it currently stands. Therefore, we invite you to submit a revised version of the manuscript that addresses the points raised during the review process.

We look forward to receiving your revised manuscript.

Kind regards,

Ashraf Elbahrawy

Academic Editor

PLOS ONE

“This work was supported by the Health, Labor, and Welfare Policy Research Grants from the Ministry of Health, Labor, and Welfare of Japan (Policy Research for Hepatitis Measures [H30-Kansei-Shitei-003]).”

“This work was supported by the Health, Labor, and Welfare Policy Research Grants from the Ministry of Health, Labor, and Welfare of Japan (Policy Research for Hepatitis Measures [H30-Kansei-Shitei-003]).”

“This work was supported by the Health, Labor, and Welfare Policy Research Grants from the Ministry of Health, Labor, and Welfare of Japan (Policy Research for Hepatitis Measures [H30-Kansei-Shitei-003]).”

5. We note that you have indicated that there are restrictions to data sharing for this study. For studies involving human research participant data or other sensitive data, we encourage authors to share de-identified or anonymized data. However, when data cannot be publicly shared for ethical reasons, we allow authors to make their data sets available upon request. For information on unacceptable data access restrictions, please see http://journals.plos.org/plosone/s/data-availability#loc-unacceptable-data-access-restrictions.

6. In the online submission form you indicate that your data is not available for proprietary reasons and have provided a contact point for accessing this data. Please note that your current contact point is a co-author on this manuscript. According to our Data Policy, the contact point must not be an author on the manuscript and must be an institutional contact, ideally not an individual. Please revise your data statement to a non-author institutional point of contact, such as a data access or ethics committee, and send this to us via return email. Please also include contact information for the third party organization, and please include the full citation of where the data can be found.

Reviewers' comments:

Reviewer's Responses to Questions

**Comments to the Author**

1. Is the manuscript technically sound, and do the data support the conclusions?

Reviewer #1: Yes

Reviewer #2: Yes

2. Has the statistical analysis been performed appropriately and rigorously? 

Reviewer #1: Yes

Reviewer #2: Yes

3. Have the authors made all data underlying the findings in their manuscript fully available?

Reviewer #1: Yes

Reviewer #2: Yes

4. Is the manuscript presented in an intelligible fashion and written in standard English?

Reviewer #1: Yes

Reviewer #2: Yes

5. Review Comments to the Author

Reviewer #1: I would like to thank you for this interesting , novel and creative work.

I just want few explained points.

1. as regard mentioned HCCs, were they the first time to be treated and assessed (i mean, not treated before).

2. In your opinion, what is the expected time to apply bio trace map , (during ablation, or short time after ablation)

3. What would happen, if we re ablated the same lesion at the same time time due to insufficient ablation.?. Would bio trace map work well?

4.Is there a difference in results, if we compare the efficacy of bio trace map in different segments of liver?

.

At the end , I am appreciating this impressive work.

Thanks for all authors.

Reviewer #2: The authors of this manuscript presented a novel Real-Time Ultrasound-based imaging analysis software for monitoring radiofrequency ablation and visualizing ablation areas. This title is worthy of investigation and the study is well organized and well structured. However, some comments are mentioned below that need more clarification, explanation, or modification.

1. The abstract of the study is better to be structured.

2. All the abbreviations in the manuscript like CT, MRI, etc. should first be mentioned in their complete forms followed by their abbreviated forms.

3. Please mention the type of study in the method section.

4. How many times each experiment was applied? Please explain.

5. Was the sample size enough to reach an accurate and robust result? Please explain.

6. Did your study involve any sort of follow-ups or not?

7. Please mention exactly the control groups that you have utilized in your study.

Wish you luck and prosperity in revising the manuscript.

6. PLOS authors have the option to publish the peer review history of their article (what does this mean?). If published, this will include your full peer review and any attached files.

Reviewer #1: **Yes: **Mohammed ElFayoumie

Reviewer #2: **Yes: **Ali Kashkooe

---

## [Author Response · Author response to Decision Letter 0]

21 Nov 2024

Point-by-point responses to the reviewers’ comments:

Reviewer 1

Generally

1. As regard mentioned HCCs, were they the first time to be treated and assessed (I mean, not treated before).

 [Response]

While some patients may have had previous treatments with radiofrequency ablation (RFA) for other lesions, the specific lesions included in this study were being treated for the first time. We have added the following to our manuscript.

Page 7, “Patients and procedures”

HCC lesions included in this study were undergoing initial treatment.

2. In your opinion, what is the expected time to apply bio trace map (during ablation, or short time after ablation).

 [Response]

We appreciate the reviewer’s comment for pointing out an important point. BioTrace map is expected to be used in real time during ablation to confirm the adequacy of the margins in a clinical practice. We revised our manuscript as follows.

Page 14, first paragraph

In clinical practice, the BioTrace map is expected to be used in real time during ablation to confirm the adequacy of the margins. This approach ensures that sufficient tissue has been ablated and helps identify any areas needing further treatment. Additionally, the BioTrace system offers advantages over CT for intra-operative monitoring, as it provides comparable performance in assessing the treatment response of radiofrequency ablation (RFA). This capability could lead to a reduction in patient radiation exposure in the future, enhancing the safety and effectiveness of the procedure.

3. What would happen, if we re ablated the same lesion at the same time due to insufficient ablation?

 [Response]

BioTrace utilizes a proprietary algorithm to analyze the temporo-spatial behavior of thermal gas bubble activity during ablation. If re-ablation is performed, BioTrace similarly analyzes the new gas bubble activity and generates an updated heat map reflecting the additional tissue damage in real time. This would allow us to visualize the extended ablation area and assess the effectiveness of the repeated ablation. 

We have added this information to our manuscript as follows.

Page 15, second paragraph

In cases of re-ablation due to insufficient ablation, BioTrace may be used to analyze the new gas bubble activity and to generate an ablation map in real time, allowing us to visualize the extended ablation area and assess the effectiveness of the repeated ablation. However, further investigation is needed to verify these observations and fully understand their implications in clinical practice.

4. Is there a difference in results, if we compare the efficacy of bio trace map in different segments of liver?

 [Response]

We conducted an ANOVA test to examine the differences among the liver segments. Although our sample size was limited, the analysis did not reveal any statistically significant differences in efficacy across the different segments. 

We have added the following to our manuscript.

Page 11, second paragraph

To address the differences in efficacy of the BioTrace map across different liver segments, we employed an analysis of variance (ANOVA) test.

Page 12, second paragraph of “Predictive accuracy of BioTrace Map for demonstrating the ablated area”

The ANOVA test did not reveal differences based on the tumor’s localized segment (p = 0.4).

Table 2

We added a column for “Tumor segment”.

Reviewer 2

1. The abstract of the study is better to be structured.

[Response]

According to the reviewer’s comment, we have structured the abstract.

2. All the abbreviations in the manuscript like CT, MRI, etc. should first be mentioned in their complete forms followed by their abbreviated forms.

[Response]

We appreciate the reviewer’s comment. In accordance with the reviewer’s comment, we have spelled out all abbreviations upon their first appearance in the manuscript.

3. Please mention the type of study in the method section.

[Response]

According to the reviewer’s comment, we have included a note specifying that the study is a prospective observational study as follows.

Page 7, “Method” section

This study is a prospective observational study, aiming to evaluate the real-time monitoring capabilities of BioTrace system.

4. How many times each experiment was applied? Please explain.

[Response]

Each experiment involving the BioTrace system was conducted once. This was due to the exploratory nature of this study. We have added this information to our manuscript as follows.

Page 11, second paragraph

Due to the exploratory nature of this study, each experiment involving the BioTrace system was conducted once.

5. Was the sample size enough to reach an accurate and robust result? Please explain.

[Response]

The sample size of 20 cases may not be sufficient to draw definitive conclusions about the robustness and accuracy of the results. This study is intended as a preliminary investigation to explore the potential of the BioTrace system, and we acknowledge that further studies with larger sample sizes are required to validate our findings and ensure their generalizability. We have added this clarification to the discussion section as follows.

Page 15, second paragraph

The limitation of this study is its small sample size of 20 cases, which may not be sufficient to draw definitive and robust conclusions regarding the efficacy and accuracy of the BioTrace system.

6. Did your study involve any sort of follow-ups or not?

[Response]

Although we are currently considering a follow-up study to further evaluate the long-term effects of the BioTrace system, the current study does not include follow-up procedures to draw conclusions regarding the efficacy and accuracy of the BioTrace system. We have included this limitation in our manuscript as follows. We have added this limitation to our manuscript as follows.

Page 15, second paragraph

Additionally, the study did not involve any follow-up procedures, which limits our ability to assess the long-term impact and outcomes of the BioTrace system.

7. Please mention exactly the control groups that you have utilized in your study.

[Response]

The current study do not include a control group. The objective of this study is to evaluate the predictive capabilities of the BioTrace system by comparing its performance against radiologist-annotated ablation zones using contrast-enhanced CT as the reference standard. 

We have added this limitation to our manuscript as follows.

Page 16, first paragraph

Finally, the study did not include a traditional control group, which limits the ability to demonstrate the added clinical value of the BioTrace system. Further studies are needed to assess the potential clinical benefits and added value of the BioTrace system in a controlled setting.

---

## [Decision Letter · Decision Letter 1]

18 Dec 2024

PONE-D-24-37385R1A Novel Real-Time Ultrasound-Based Imaging Analysis Software for Monitoring Radiofrequency Ablation and Visualizing Ablation AreasPLOS ONE

Dear Dr. Sato,

Thank you for submitting your manuscript to PLOS ONE. After careful consideration, we feel that it has merit but does not fully meet PLOS ONE’s publication criteria as it currently stands. Therefore, we invite you to submit a revised version of the manuscript that addresses the points raised during the review process.

We look forward to receiving your revised manuscript.

Kind regards,

Ashraf Elbahrawy

Academic Editor

PLOS ONE

Reviewers' comments:

Reviewer's Responses to Questions

**Comments to the Author**

1. If the authors have adequately addressed your comments raised in a previous round of review and you feel that this manuscript is now acceptable for publication, you may indicate that here to bypass the “Comments to the Author” section, enter your conflict of interest statement in the “Confidential to Editor” section, and submit your "Accept" recommendation.

Reviewer #1: All comments have been addressed

Reviewer #3: All comments have been addressed

2. Is the manuscript technically sound, and do the data support the conclusions?

Reviewer #1: Yes

Reviewer #3: Partly

3. Has the statistical analysis been performed appropriately and rigorously? 

Reviewer #1: Yes

Reviewer #3: Yes

4. Have the authors made all data underlying the findings in their manuscript fully available?

Reviewer #1: Yes

Reviewer #3: Yes

5. Is the manuscript presented in an intelligible fashion and written in standard English?

Reviewer #1: Yes

Reviewer #3: Yes

6. Review Comments to the Author

Reviewer #1: (No Response)

Reviewer #3: In the current study you compared performance of BTM( pre ablation prediction) with CECT 24 hours after ablation, please mention this in a clear statement in methodology section. In addition, BTM system is not activated during the procedure in the current study. So I think the title should be modified express the work in the study.

Please revise the following:

((A Novel Real-Time Ultrasound-Based Imaging Analysis Software for Monitoring Radiofrequency Ablation and Visualizing Ablation Areas))

In the current study you said in page 15” Although the current system has been designed for real-time monitoring, it was not activated during the procedure in this study.

Page 14” It provides comparable performance in assessing the treatment response of radiofrequency ablation (RFA).”

Unfortunately, the study design is not compatible with prospective observational study! You compared performance of BTM with CECT 24 hours after ablation in the same group.

What is/are the extra benefit/s expected from BTM in contrast to ablation under the guidance of contrast-enhanced ultrasonography? Both are ultrasound based and real time technique!

Page 8, you mentioned “The dynamics of microbubbles, reflecting the tissue’s response to heat, varies according to the applied temperature” I think there are multiple variables other than temperature should be considered e.g., biological nature of lesion, pathological background of liver. I mean HCC on top of chronic hepatitis, cirrhosis or viral VS metabolic liver disease, fibrotic load within the liver,……

7. PLOS authors have the option to publish the peer review history of their article (what does this mean?). If published, this will include your full peer review and any attached files.

Reviewer #1: **Yes: **Mohammed Elfayoumie

Reviewer #3: **Yes: **Ali Madian

---

## [Author Response · Author response to Decision Letter 1]

26 Dec 2024

Point-by-point responses to the reviewers’ comments:

Reviewer 3

1. In the current study, you compared performance of BTM (pre ablation prediction) with CECT 24 hours after ablation, please mention this in a clear statement in methodology section. In addition, BTM system is not activated during the procedure in the current study. So I think the title should be modified express the work in this study.

Please revise the following:

((A Novel Real-Time Ultrasound-Based Imaging Analysis Software for Monitoring Radiofrequency Ablation and Visualizing Ablation Areas))

In the current study you said in page 15” Although the current system has been designed for real-time monitoring, it was not activated during the procedure in this study.

Page 14” It provides comparable performance in assessing the treatment response of radiofrequency ablation (RFA).”

Unfortunately, the study design is not compatible with prospective observational study! You compared performance of BTM with CECT 24 hours after ablation in the same group.

 [Response]

We agree that the methodology and title should be clarified to better reflect the nature of our study. 

We modified the title of our manuscript as follows

Page 1, Title 

Retrospective Evaluation of a Novel Ultrasound-Based Imaging Analysis Software for Predicting Radiofrequency Ablation Areas

We have added the following statement in the Method section under "Performance evaluation of BioTrace and statistical analysis” as follows.

Page 7, “Patients and procedures”

This study is a retrospective analysis of prospectively collected ultrasound imaging data, aiming to evaluate the predictive capabilities of the BioTrace system.

Page 9, “Performance evaluation of BioTrace and statistical analysis”

In this study, we retrospectively analyzed the ablation areas by comparing the BioTrace prediction generated from ultrasound imaging data collected during the RFA procedure with the actual ablation zones confirmed by CECT obtained 24 hours post-treatment, which is considered the gold standard. While the BTM system is designed for real-time monitoring, it was not activated during the procedures in this study. Instead, we retrospectively applied the BioTrace algorithm to the prospectively collected ultrasound imaging videos to generate the ablation prediction maps.

Regarding the description on page 14 and page 15 (in the pre revise version), we have addressed the issues and revised the manuscript as follows.

Page 15, first paragraph

Additionally, the BioTrace system demonstrated comparable performance to CT in assessing ablation zones in this retrospective analysis, suggesting potential advantages for future intra-operative monitoring applications.

Page 16, second paragraph

While the BioTrace system is designed for real-time monitoring, this validation study focused on retrospective analysis of prospectively collected ultrasound imaging data. The real-time monitoring capabilities will be evaluated in future prospective clinical trials.

2. What is/are the extra benefit/s expected from BTM in contrast to ablation under the guidance of contrast-enhanced ultrasonography? Both are ultrasound based and real time technique!

 [Response]

We appreciated the reviewer’s important question regarding the advantages of BioTrace (BTM) compared to contrast-enhanced ultrasonography. The BioTrace system offers several distinct advantages over contrast-enhanced ultrasonography (CEUS). First, it minimizes the resources required for contrast agent preparation and administration. Second, unlike CEUS which requires low acoustic pressure to maintain microbubble stability, BioTrace can utilize optimal ultrasound settings to maintain high-quality B-mode imaging throughout the procedure. This allows for better visualization of the target area and surrounding structures during ablation.

We have added the current information to our manuscript as follows.

Page 14, third paragraph

The BioTrace system offers several distinct advantages over contrast-enhanced ultrasonography. First, it minimizes the resources required for contrast agent preparation and administration. Second, unlike CEUS which requires low acoustic pressure to maintain microbubble stability, BioTrace can utilize optimal ultrasound settings to maintain high-quality B-mode imaging throughout the procedure. This allows for better visualization of the target area and surrounding structures during ablation.

3. Page 8, you mentioned “The dynamics of microbubbles, reflecting the tissue’s response to heat, varies according to the applied temperature” I think there are multiple variables other than temperature should be considered e.g., biological nature of lesion, pathological background of liver. I mean HCC on top of chronic hepatitis, cirrhosis or viral VS metabolic liver disease, fibrotic load within the liver,……

 [Response]

We understand the reviewer’s comment. We have revised the manuscript to better reflect the complexity of these factors influencing the ablation process as follows.

Page 8, second paragraph

The dynamics of microbubbles reflect a complex interaction of multiple variables during thermal ablation. While temperature is a primary factor, the microbubble behavior is also influenced by: (1) the biological nature of the lesion, (2) the underlying pathological condition of the liver (such as chronic hepatitis, cirrhosis, or metabolic liver disease), (3) the degree of liver fibrosis, (4) local tissue perfusion characteristics, and (5) the applied temperature. These variables collectively affect the formation and the dynamic behavior of microbubbles during thermal ablation.

We also added the text to emphasize the need for further investigation across different liver conditions to better understand these complex interactions as follows.

Page 16, second paragraph

Furthermore, the influence of various liver pathologies and tissue characteristics on microbubble dynamics requires additional investigation. Future studies should evaluate the system's performance across different pathological conditions, including chronic hepatitis, cirrhosis, and varying degrees of liver fibrosis, to better understand how these factors affect the accuracy of ablation zone prediction.

---

## [Editor Report · Decision Letter 2]

29 Dec 2024

Retrospective Evaluation of a Novel Ultrasound-Based Imaging Analysis Software for Predicting Radiofrequency Ablation Areas

PONE-D-24-37385R2

Dear Dr. Sato,

We’re pleased to inform you that your manuscript has been judged scientifically suitable for publication and will be formally accepted for publication once it meets all outstanding technical requirements.

Kind regards,

Ashraf Elbahrawy

Academic Editor

PLOS ONE
---

## [Editor Report · Acceptance letter]

7 Jan 2025

PONE-D-24-37385R2 

PLOS ONE

Dear Dr. Tateishi, 

I'm pleased to inform you that your manuscript has been deemed suitable for publication in PLOS ONE. Congratulations! Your manuscript is now being handed over to our production team.

Kind regards, 

on behalf of

Prof. Ashraf Elbahrawy 

Academic Editor

PLOS ONE